:ᴗ: PLOS ONE

# Exploring the basis of 2-propenyl and 3-butenyl glucosinolate synthesis by QTL mapping and RNA-sequencing in *Brassica juncea*

**Aimal Nawaz Khattak**[☉]**, Tianya Wang**[☉]**, Kunjiang Yu, Renqin Yang, Wei Wan, Botao Ye, Entang Tian**[ID] *

Oil Crops Research Institute of Guizhou University, Agricultural College of Guizhou University, Guizhou University, Guiyang, China

[☉] These authors contributed equally to this work.
* erictian121@163.com

**Data Availability Statement:** All relevant data are within the paper and its Supporting Information files.

## Abstract

*Brassica juncea* is used as a condiment, as vegetables and as an oilseed crop, especially in semiarid areas. In the present study, we constructed a genetic map using one recombinant inbred line (RIL) of *B. juncea*. A total of 304 ILP (intron length polymorphism) markers were mapped to 18 linkage groups designated LG01-LG18 in *B. juncea*. The constructed map covered a total genetic length of 1671.13 cM with an average marker interval of 5.50 cM. The QTLs for 2-propenyl glucosinolates (GSLs) colocalized with the QTLs for 3-butenyl GSLs between At1g26180 and BnapPIP1580 on LG08 in the field experiments of 2016 and 2017. These QTLs accounted for an average of 42.3% and 42.6% phenotypic variation for 2-propenyl and 3-butenyl GSLs, respectively. Furthermore, the Illumina RNA-sequencing technique was used to excavate the genes responsible for the synthesis of GSLs in the siliques of the parental lines of the RIL mapping population, because the bulk of the seed GSLs might originate from the siliques. Comparative analysis and annotation by gene ontology (GO) and kyoto encyclopedia of genes and genomes (KEGG) revealed that 324 genes were involved in GSL metabolism, among which only 24 transcripts were differentially expressed genes (DEGs). Among those DEGs, 15 genes were involved in the biosynthesis and transport of aliphatic GSLs, and their expression patterns were further validated by qRT-PCR analysis. Joint QTL mapping and RNA-sequencing analyses reveal one candidate gene of *IIL1* (LOC106416451) for GSL metabolism in *B. juncea*. These results will be helpful for further fine mapping, gene cloning and genetic mechanisms of 2-propenyl and 3-butenyl GSLs in *B. juncea*.

## Introduction

*Brassica juncea* (AABB, 2n = 36) is an important allotetraploid species that originated from interspecific hybridization between *B. rapa* (AA, 2n = 20) and *B. nigra* (BB, 2n = 16) followed

**Funding:** This work was funded by the National Natural Science Foundation of China (Grant No. 31560422), Agricultural Science and Technology Support Program of Guizhou Province (Qiankehe zhicheng No. [2019]2396), Science and Technology Foundation of Guizhou Province of China (Grant No. Qiankehe J zi [2015]2052), Scientific Research Foundation for Returned Scholars, Ministry of Education of China (Grant No. Jiaowaisiliu [2015]1098), Foundation of Guizhou University (Grant No. Guidarenjihezi [2014]14), Construction, Construction Program of Biology First-class Discipline in Guizhou (Grant No. GNYL [2017]009). The funders had no role in study design, data collection and analysis, decision to publish, or preparation of the manuscript.

**Competing interests:** The authors have declared that no competing interests exist.

by chromosome doubling in the natural environment. The crop exhibited better drought and heat tolerance, disease resistance, insect resistance and shattering resistance than *B. napus* [1–7]. In addition to its use as a condiment in Canada and China and as a vegetable in China, great efforts have been made to develop *B. juncea* as an alternative oilseed crop, especially in the semiarid areas. Canola-quality *B. juncea* with less than 2% erucic acid in the seed oil and less than 30-uM GSLs/g of the deoiled cake was developed through a cross between a zero erucic *B. juncea* line and a low GSLs *B. juncea* line [8]. Hybrid breeding has been successfully used to enhance the yield potential in canola *B. napus*. In *B. juncea*, the Ogura cytoplasmic male sterility (cms) and its restorer gene (*Rfo*) were introduced for heterosis utilization [9,10]. Then, the *B. juncea* Ogura cms restorer line (*RfoRfo*) was improved with drastically reduced linkage drag, good seed set and high agronomic performance by hybridization with resynthesized *B. juncea* lines and subsequent molecular marker-assisted selection in *B. juncea* [11].

Glucosinolates (GSLs) were first discovered in mustard seeds during an exploration of the chemical origin of their sharp taste in the 17th century. To date at least 120 different GSLs have been identified in sixteen families of dicotyledonous angiosperms [12]. In the Brassicaceae family, GSLs are the major secondary metabolites and could be synthesized in all species of this family by a three-part biosynthetic pathway from methionine, tryptophan and phenylalanine [13–15]. GSLs are mainly divided into aliphatic, indolic and benzyl GSLs in *Brassica* species. Most of the tissues, such as rosette leaves, roots, seeds, inflorescences, contain GSLs. The GSL contents of different tissues are not entirely synthesized locally. The transport of GSLs was suggested more than 40 years ago through a number of studies that indicated that GSLs are produced in maternal tissue and subsequently transported to the seed [16,17]. This could explain why the GSL profiles of hybrid seeds were similar to those of the maternal plants instead of being intermediate between those of the maternal and paternal plants [16]. As the closest organ to the seeds, the silique is the only organ to produce all the GSLs found in the seed [17–19]. Thus, the siliques might be the most important source of seed GSLs.

In seeds of the Brassicaceae, aliphatic GSLs were the major type of GSLs, while 2-propenyl and 3-butenyl GSLs were the major types of aliphatic GSLs. Previous studies have shown that the European type of *B. juncea* mainly contains 2-propenyl GSLs and Indian types contain both 2-propenyl and 3-butenyl GSLs [20–22]. The inheritance of 2-propenyl and 3-butenyl GSLs was studied in $F_1$, $F_2$ and backcrossing populations of *B. juncea*, indicating that these two characteristics had maternal effects and might be controlled by multiple additive alleles at the same loci [21]. Using bulked segregant analysis, one ISSR marker was found to be tightly linked to high 2-propenyl GSLs in *B. juncea* and converted to a SCAR marker [23]. A total of 17 metabolic QTLs for the genetic control of 2-propenyl and 3-butenyl GSLs were identified on LG2, LG3, LG4, LG5, LG6, LG7, LG8 and LG9 in *B. oleracea*, among which 12 regulated 2-propenyl and 3-butenyl GSLs at the same time, 2 were specific for 2-propenyl GSLs and 3 were specific for 2-propenyl GSLs [24]. In *B. napus* seeds, 3 QTLs on A9 and 1 QTL on C2 for the genetic regulation of 3-butenyl GSLs were detected, and each could explain a phenotypic variation between 5.0% and 14.8% [25].

ILP markers utilize variations in intron sequences and are the most easily recognizable type of marker, as they can be detected by PCR with primers designed for the exons flanking the target intron [26]. Furthermore, ILP markers are unique because they are gene-specific, codominant, hypervariable, neutral, convenient and reliable [26,27]. These markers have been used for genetic analysis in many species, such as rice, yellow mustard, foxtail millet, maize, tomato, *B. juncea*, *B. rapa* and Arabidopsis [26–32]. In the present study, we successfully used PCR-based ILP markers for the development of a genetic map based on one RIL mapping population in *B. juncea*. Furthermore, we detected one novel major QTL for 2-propenyl and 3-butenyl GSLs on LG08 of the *B. juncea* genome. In addition, we also attempted to explore

the mechanism resulting in the variation in seed GSL contents by RNA-sequencing of siliques from the parental lines of the RIL mapping population.

## Materials and methods

### Plant materials and field trial

Parents G266 and G302 are DH lines. The seeds of the G266 line had low 2-propenyl and high 3-butenyl GSL contents, while G302 had high 2-propenyl and low 3-butenyl GSL contents. The RILs produced by G266×G302 displayed great variation in agronomic traits, such as flowering time and number of seeds per silique as presented in our earlier study [33]. Three replicates for each of the parental lines G266 and G302, their $F_1$ and 167 $F_6$ RILs were planted on the farm of Guizhou University, Guiyang, China in 2016 and 2017. The design of the trial was a randomized complete block. Each plot consisted of two rows with one size of 3.66 $m^2$ (3 m×1.22 m). Five grams of seeds from three plants in each plot were harvested at maturity and analyzed for 2-propenyl and low 3-butenyl GSL contents. The average 2-propenyl and low 3-butenyl GSL contents of the three replicates were used for QTL analysis.

### DNA extraction and polymerase chain reaction (PCR)

Genomic DNA was extracted from young leaves of the parental lines, $F_1$ and 167 $F_6$ RILs using the modified sodium dodecyl sulfate method [34]. PCR of the ILP markers was carried out according to our previous studies [11,35]. Each PCR (20 μl) contained 1× standard PCR buffer (NEB), 1 U of Taq polymerase (NEB), 0.25 μM forward primer, 0.25 μM reverse primer, 100 μM each dNTP and 50 ng of genomic DNA in a total volume of 20 μL. The PCR amplification consisted of an initial denaturation at 94˚C for 5 min; 35 cycles consisting of 94˚C (45 sec); 55˚C (45 sec) and 72˚C (1 min); followed by termination at 72˚C for 7 min. All PCR products were analyzed by electrophoresis in 2% agarose gels in 1× tri-acetate-ethylene diaminetetra acetic acid buffer. Gels were visualized by staining in ethidium bromide and photographed on a digital gel documentation system.

### Construction of genetic linkage map and QTL analysis

The genetic linkage map of *B. juncea* was constructed by using JoinMap 4.0 software at LOD≥4.0 [36]. Recombination frequencies were converted to map distances in cm using the Kosambi mapping function and the genetic map was drawn with MapChart software [37]. QTL analysis of 2-propenyl and 3-butenyl GSL contents was performed using the interval mapping method of MapQTL 6.0 software [38]. A permutation test (1,000 replications) was used to determine the significance level for LOD with a genome-wide probability of $p < 0.05$.

### Glucosinolate component analysis

The 2-propenyl and 3-butenyl GSL contents of the mature seeds from each plot were analyzed following published methods [39,40] with minor modifications. Each seed sample was crushed and 200 mg of each sample was extracted twice with 2 ml boiling 70% methanol. The concentration of GSLs in the seeds was determined by high-performance liquid chromatography (Waters 2487/600/717) using the ISO9167-1 (1992) standard method.

### RNA extraction, preparation, sequencing and data analysis

Total RNA (2 μg) was extracted from fresh seed coats of 20 DAP (days after pollination) siliques of three independent plants for each of the parental lines G266 and G302 using the TRIzol kit (Invitrogen, Carlsbad, CA), according to the manufacturer's instructions. The RNA

purity was checked using the Kaiao K5500®Spectrophotometer (Kaiao, Beijing, China), and the RNA integrity and concentration were assessed using the RNA Nano 6000 Assay Kit for the Bioanalyzer 2100 system (Agilent Technologies, CA, USA). Then, the six RNA samples were sent to the ANOROAD GENOME company (http://www.genome.cn/) for the construction of cDNA libraries and Illumina deep sequencing according to the paper of Wang et al. [41]. The raw RNA-sequencing data were filtered by a Perl script, following the steps of Wu et al. [42].

## Identification and annotation of differentially expressed genes (DEGs)

DESeq2 v1.6.3 was designed for differential gene expression analysis between two samples with three biological replicates under the theoretical basis obeys the hypothesis of negative binomial distribution for the value of count. The p-value was corrected by the BH method. Genes with q≤0.05 and |log2_ratio|≥1 were identified as differentially expressed genes (DEGs) [43]. The DEGs obtained were further annotated with Gene Ontology (GO, http://geneontology.org/) and analyzed by KEGG (Kyoto Encyclopedia of Genes and Genomes, http://www.kegg.jp/) [44,45]. The GO enrichment of DEGs was implemented by the hypergeometric test, in which the p-value is calculated and adjusted to produce the q-value, and the data background is the genes in the whole genome. GO terms with q<0.05 were considered to be significantly enriched. GO enrichment analysis was used to determine the biological functions of the DEGs. KEGG is a database resource containing a collection of manually drawn pathway maps representing our knowledge of molecular interaction and reaction networks. The KEGG enrichment of the DEGs was determined by the hypergeometric test, in which p-value was adjusted by multiple comparisons to produce the q-value. KEGG terms with q<0.05 were considered to be significantly enriched.

## Quantitative real time-PCR (qRT-PCR) analysis

Quantitative real-time PCR (qRT-PCR) was used to verify the transcript levels of the RNA-Seq results. Total RNA was extracted using the TRIzol kit (Invitrogen), according to the manufacturer's instructions. Then, the cDNA was synthesized by reverse transcription using Prime-Script RT reagent kits with gDNA Eraser (Takara, Dalian, China) according to the manufacturer's instructions. Sixteen gene-specific primers for qRT-PCR were designed based on reference unigene sequences randomly chosen from the DEGs using Primer Premier 5.0. Real-time PCR was conducted using SsoAdvanced[TM] Universal SYBRGreen Supermix (Hercules, CA) in a typical 20 µl PCR mixture. The 20 µl mixture contained 10 µl SYBR Green Supermix (2×), 0.4 µl reverse and forward primers (10 µM), 2 µl (100 ng) template cDNA, and 7.2 µl ddH$_2$O. The qRT-PCR conditions were 95°C for 2 min, followed by 40 cycles of 95°C for 10 s (denaturation), followed by 60°C for 20 s (annealing and extension). The $2^{-\Delta\Delta Ct}$ algorithm was used to calculate the relative level of gene expression. The *β-actin* gene was used as the internal control, and the T399 samples served as the control. All qRT-PCR were performed with three biological replicates, and run on a Bio-Rad CFX96 Real Time System (Bio-Rad, Hercules, CA, USA).

## Results

### Polymorphism between the parental lines G266 and G302

A total of 1,272 ILP primers, 284 from *Arabidopsis thaliana* [32], 745 from *B. napus* and 243 from *B. rapa* available in the Potential Intron Polymorphism (PIP) database [27], were used to screen the parental lines G266 and G302 for polymorphic primers. Of the 1,272 ILP primers,

306 (24.1%) generated clear and scorable polymorphic bands between the parental lines varying in size from 150 to 1250 bp. Among the 306 polymorphic primers, 266 (86.9%) amplified one locus, 35 (12.4%) produced two loci, 4 (1.4%, At4g11790, At1g07980, PIP1848 and At3g52990) produced three loci, 2 (0.7%, PIP1202 and PIPR68) revealed four loci and one (0.4%, At1g72890) revealed five loci. In summary, 359 polymorphic markers were amplified by 306 polymorphic primers, including 231 dominant ones and 128 codominant ones. The 359 polymorphic markers were used to construct the linkage map with the RIL population of G266×G302 in *Brassica juncea*.

### Construction of one genetic linkage map

A total of 304 polymorphic loci of the 359 polymorphic markers (84.7%) were mapped on 18 linkage groups and covered a genetic length of 1671.13 centiMorgans (cM) with an average marker interval of 5.50 cm (Table 1 and Figs 1 and 2). The linkage groups were designated as LG01-LG18.

The map lengths of the 18 linkage groups ranged from 54.41 cM for LG16 to 180.48 cM for LG08 with an average of 92.843 cM. The marker interval ranged from 0.00 cM to 37.39 cM with an average of 5.75 cM. LG03, LG08, LG14 and LG18 had map lengths longer than 100 cM, ranging from 110.44 cM to 180.48 cM. LG18 had the largest average marker interval of 11.04 cM. LG08 had the longest map length of 180.48 cM and the most ILP markers (35 markers). LG07, LG15 and LG17 had similar map lengths ranging from 93.83 cM to 97.63 cM. LG01, LG04, LG09, LG11 and LG12 had similar map lengths ranging from 80.25 cM to 88.37 cM. LG04 had the largest island without markers (37.39 cM). LG10, LG06 and LG13 had similar long map lengths ranging from 75.00 cM to 78.53 cM. LG02, LG05 and LG16 had similar long map lengths ranging from 54.41 cM to 64.38 cM. LG05 had the shortest map length of 54.41 cM and the smallest average marker interval of 1.92 cM.

**Table 1. Characterization of the 18 linkage groups in *Brassica juncea*.**

| Linkage Group | Map Length (cM) | Marker interval (cM) | | | No. of markers |
| --- | --- | --- | --- | --- | --- |
| | | Average | Max Distance (cM) | Min Distance (cM) | |
| LG01 | 88.37 | 5.52 | 24.19 | 0.06 | 17 |
| LG02 | 64.38 | 5.36 | 13.79 | 0.35 | 13 |
| LG03 | 134.25 | 6.39 | 19.30 | 0.01 | 22 |
| LG04 | 82.05 | 8.21 | 37.39 | 0.33 | 11 |
| LG05 | 61.31 | 1.92 | 20.96 | 0.22 | 33 |
| LG06 | 77.10 | 7.71 | 33.53 | 0.00 | 10 |
| LG07 | 93.83 | 11.73 | 25.79 | 0.77 | 9 |
| LG08 | 180.48 | 5.31 | 15.07 | 0.00 | 35 |
| LG09 | 80.35 | 3.21 | 16.21 | 0.19 | 26 |
| LG10 | 75.00 | 7.50 | 16.21 | 0.83 | 11 |
| LG11 | 83.13 | 7.56 | 13.69 | 0.74 | 12 |
| LG12 | 80.25 | 10.03 | 16.62 | 5.10 | 9 |
| LG13 | 78.53 | 8.73 | 16.85 | 2.77 | 10 |
| LG14 | 135.16 | 5.88 | 14.43 | 0.01 | 24 |
| LG15 | 94.46 | 4.11 | 24.01 | 0.04 | 24 |
| LG16 | 54.41 | 4.53 | 13.02 | 0.60 | 13 |
| LG17 | 97.63 | 7.51 | 20.67 | 0.10 | 14 |
| LG18 | 110.44 | 11.04 | 26.26 | 0.62 | 11 |
| Total | 1671.13 | - | - | - | 304 |

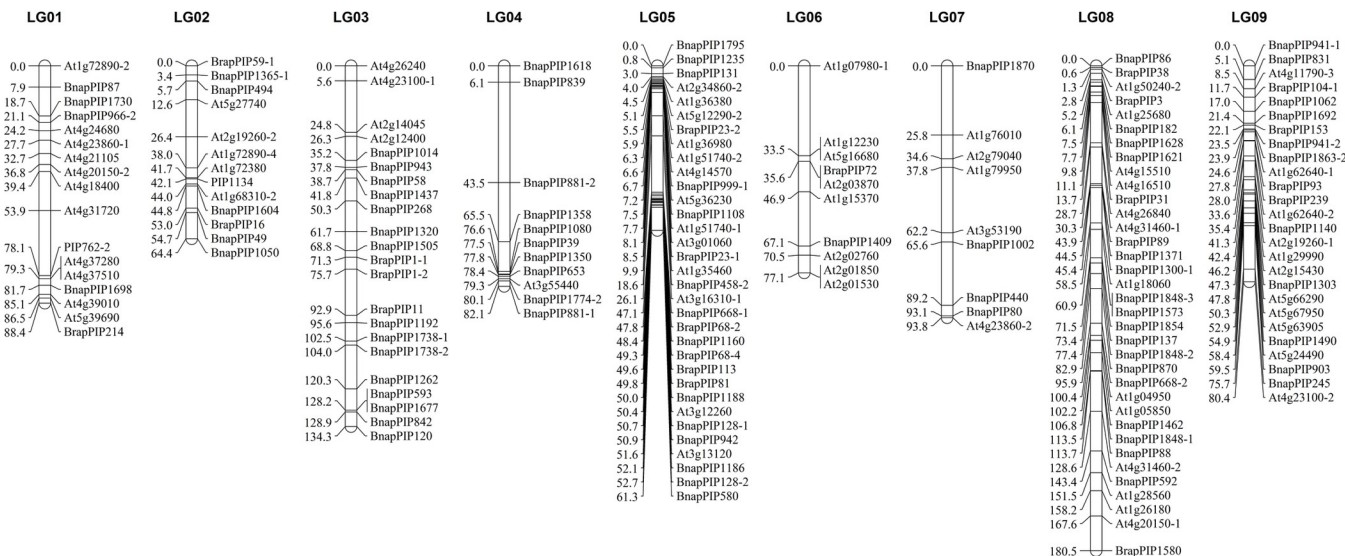

**Fig 1. The 9 linkage groups from LG01 to LG09 in *Brassica juncea*.** For each linkage group, the ILP markers were shown on the right side and the marker position in centiMorgan on the left side.

## QTL mapping of 2-propenyl and 3-butenyl glucosinolate contents

The 167 RILs, their parental lines G266 and G302, and the $F_1$, were grown in the field with three replications. These lines were grown at Guiyang and distributed normally in 2016 and 2017 (Fig 3). No significant difference across the two years for 2-propenyl and 3-butenyl GSL contents was detected (p = 0.594 and p = 0.888, respectively). The 2-propenyl GSL contents was significantly negatively correlated with the 3-butenyl GSL contents in 2016 (r = -0.920, p = 0.000) and 2017 (r = -0.914, p = 0.000), respectively. The parents of the population differed in 2-propenyl GSL contents, with mean values of 17.29 μmol/g and 180.90 μmol/g for G266 and G302, respectively (Table 2). The mean 2-propenyl GSL contents of $F_1$ was 67.34 μmol/g, which was closer to that of the female parent and slightly higher than the mean value of the RIL mapping population (63.45 μmol/g) (Table 2). The range of 2-propenyl GSL contents in the RILs was 10.70~214.36 μmol/g in 2016 and 9.59~215.60 μmol/g in 2017 (Table 2). The

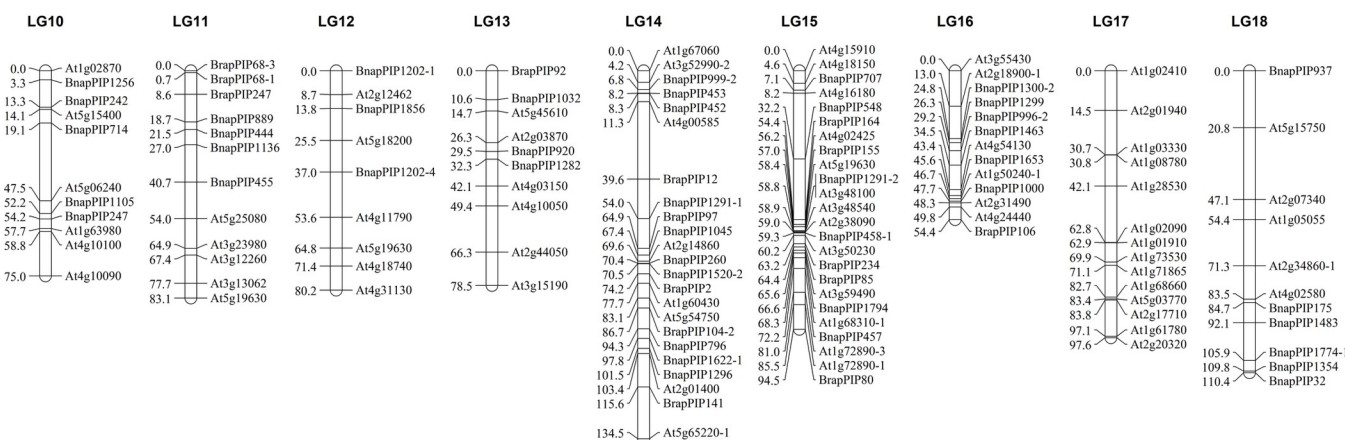

**Fig 2. The 9 linkage groups from LG10 to LG18 in *Brassica juncea*.** For each linkage group, the ILP markers were shown on the right side and the marker position in centiMorgan on the left side.

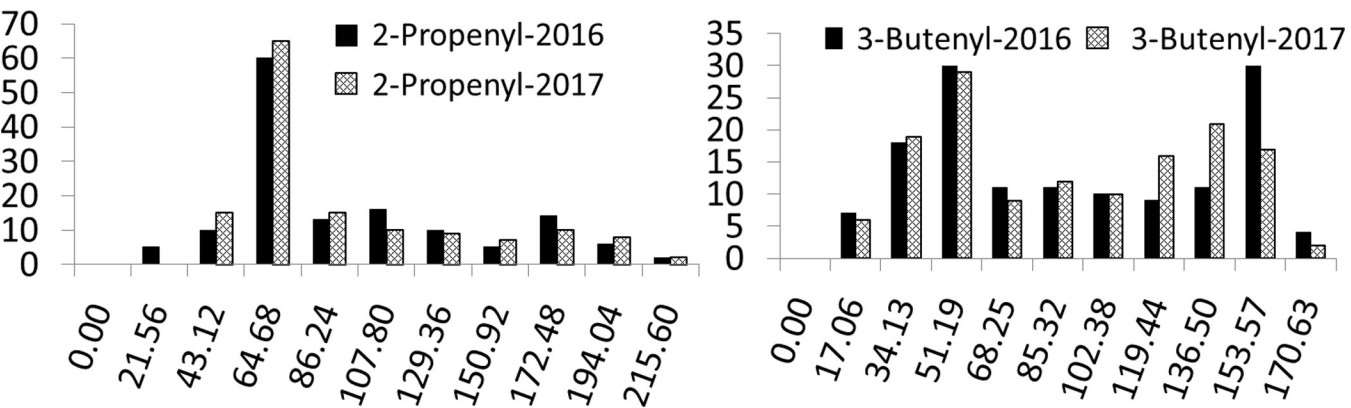

**Fig 3. Frequency distribution of 2-propenyl and 3-butenyl glucosinolate content of the recombinant inbred lines (RILs) in 2016 and 2017.** Y-axis: the number of the corresponding recombinant inbred lines; X-axis: the glucosinolate content of 2-propenyl (left) and 3-butenyl (right) respectively (u mol/g).

parents of the population differed in 3-butenyl GSL contents, with mean values of 86.98 μmol/g and 15.80 μmol/g for G266 and G302, respectively (Table 2). The mean 3-butenyl GSL contents of F$_1$ was 63.80 μmol/g, more similar to that of the female parent and slightly lower than the mean value in the RIL mapping population (75.98 μmol/g) (Table 2). The range of 3-butenyl GSL contents of the RILs was 8.71~170.63 u mol/g in 2016, and 5.82~166.04 μmol/g in 2017 (Table 2).

QTL analysis was performed for 2-propenyl (SIN, sinigrin) and 3-butenyl (GNA, gluconapin) GSL contents in 2016 and 2017 respectively. Two major QTLs of SIN-2016 and SIN-2017 for 2-propenyl GSL contents (LOD = 16.46 and 16.64) were colocalized between At1g26180 and BnapPIP1580 on LG08, accounting for 42.3% and 42.6% of the total variation in 2016 and 2017, respectively (Table 3 and Fig 4). Another two QTLs for 3-butenyl GSL contents were detected and colocalized with GNA-2016 and GNA-2017 between At1g26180 and BnapPIP1580 on LG08 (Table 3 and Fig 4). The two QTLs for 3-butenyl GSL contents explained 31% and 38.4% of the total variation in 2016 and 2017 respectively (Table 3). All these QTLs were mapped to a region adjacent to the ILP marker At4g20150-1 on LG08 (Table 3 and Fig 4).

## Synteny relationships between LG08 and A08 of *B. juncea* and *B. rapa*

LG08 contained a total of 35 ILP markers. Among these, 13 (37.1%) were developed from the single-copy genes of *A. thaliana*, and 17 and 5 were developed from the unique

**Table 2. 2-propenyl and 3-butenyl GSL content of the parental lines G266 and G302, F$_1$ seeds of G266×G302 and the RIL mapping population in 2016 and 2017.**

| Traits (u mol/g) | Years | Parents | | F$_1$ | RILs | |
|---|---|---|---|---|---|---|
| | | G266 | G302 | | Mean | Range |
| 2-Propenyl | 2016 | 19.14±0.21[a] | 181.53±7.32 | 73.50±1.12 | 65.45 | 10.70~214.36 |
| | 2017 | 15.43±0.60 | 180.27±11.51 | 61.18±1.91 | 61.44 | 9.59~215.60 |
| | mean | 17.29 | 180.90 | 67.34 | 63.45 | − |
| 3-Butenyl | 2016 | 86.79±1.00 | 20.68±0.16 | 57.66±0.89 | 76.42 | 8.71~170.63 |
| | 2017 | 87.17±1.15 | 10.91±0.63 | 69.93±1.20 | 75.53 | 5.82~166.04 |
| | mean | 86.98 | 15.80 | 63.80 | 75.98 | |

[a]: standard deviation.

**Table 3. QTLs for 2-propenyl and 3-Butenyl GSL components in the RIL mapping population derived from the cross of G266 and G302 in 2016 and 2017.**

| Name of the QTLs[a] | Chromosome | Peak Position | LOD[b] | R²,%[c] | Additive effect (µmol/g seed) | Nearest ILP and its interval to the peak |
|---|---|---|---|---|---|---|
| SIN-2016 | LG08 | 166.21 | 16.46 | 42.3 | 6.09 | At4g20150-1,1.37 |
| SIN-2017 | LG08 | 164.21 | 16.64 | 42.6 | 6.25 | At4g20150-1,3.37 |
| GNA-2016 | LG08 | 165.21 | 11.14 | 31.0 | 6.63 | At4g20150-1,2.37 |
| GNA-2017 | LG08 | 165.21 | 14.53 | 38.4 | 0.10 | At4g20150-1,2.37 |

[a] SIN: sinigrin or 2-propenyl GSL; BUT: gluconapin or 3-Butenyl GSL; 2016: the results are from the field experiment in 2016; 2017: the results are from the field experiment in 2017.

[b] LOD: logarithm of the odds score for QTLs calculated by composite interval mapping.

[c] R²: the phenotypic variation explained by a QTL in percentage.

transcript fragments of *B. napus* and *B. rapa*, respectively [27]. To validate the exact linkage group of LG08 with QTLs for 2-propenyl and 3-butenyl GSL contents, a synteny analysis was performed. Eight of the 13 *A. thaliana* markers prefixed "At" on LG08 showed synteny between LG08 and A08 of *B. juncea* (Fig 4). Furthermore, the sequences of the unique transcript fragments for developing the 22 *B. napus* and *B. rapa* markers were used to blast against the *Brassica rapa* genome (*Brassica rapa* cultivar Chiifu, Brapa_1.0) in NCBI (https://www.ncbi.nlm.nih.gov/), and 18 (81.8%) were mapped to the A08 chromosome (S1 Table). The synteny analysis indicated that LG08 was exactly chromosome A08 of the A genome.

## Transcriptome analysis of the parental siliques as potential

**GSL source for seeds.** To construct a de novo transcriptome database, three RNA libraries were generated for each of G266 and G302 lines through Illumina sequencing. A total of 139,197,152 and 138,957,982 raw reads were generated from the G266 and G302 libraries, respectively (S2 Table). After removing low quality reads, adapter polluted reads and higher N contents (>5%) reads, a total of 134,953,284 (96.95%, T399) and 134,832,452 (97.03%, T085) clean reads were obtained (S2 Table). After filtering out the genes that contained only one exon or encoded short peptide chains (<50 amino acid residues), a total of 81,826 transcripts were revealed by blasting the reference genome using DESeq2 (v1.6.3). To functionally annotate those transcripts, the 81,826 transcripts were blasted in search of Gene Ontology (GO) and Kyoto Encyclopedia of Genes and Genomes (KEGG). Finally, 8,1694 transcripts (92.64%) were successfully annotated by GO and KEGG, and 324 ones of these transcripts were involved in GSL metabolism (S3 Table).

To identify differentially expressed genes (DEGs) involved in the GSL metabolism of siliques between G266 and G302. A rigorous algorithm with a threshold of "FDR$\leq$0.05 and |Log$_2$Ratio|$\geq$1" was developed and used as thresholds to judge the significance of differences in transcript abundance. It was found that only 24 transcripts were DEGs related to GSL metabolism. Among those DEGs, 15 genes are involved in the biosynthesis and transport of aliphatic GSLs (Table 4), and 5 and 4 genes are involved in the biosynthesis of indolic GSLs and the breakdown of GSLs, respectively (S4 Table).

To verify the expression of these transcripts detected by RNA-Seq, the 15 candidate genes involved in the aliphatic GSL metabolism pathway were randomly chosen for validation by qRT-PCR. Detailed information about the chosen DEGs and primers is listed in S5 Table. The data obtained by qRT-PCR were consistent with the RNA-Seq results (Fig 5), suggesting the reliability of the transcriptome database.

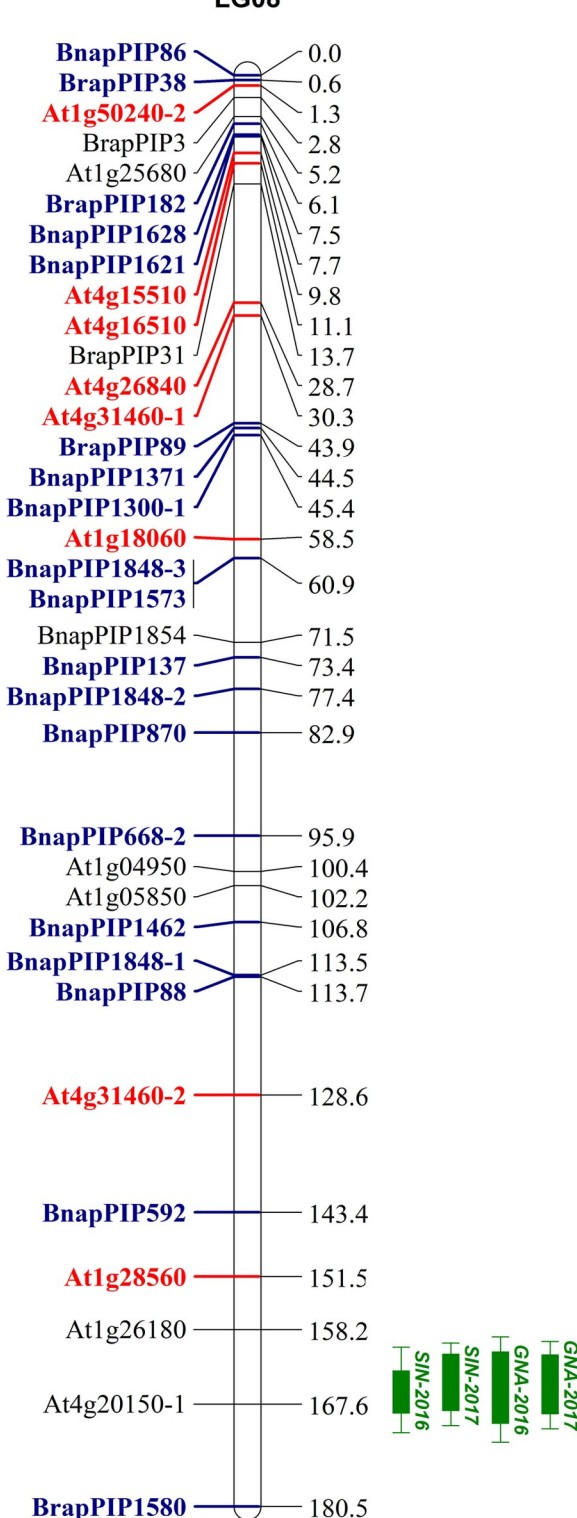

**Fig 4. QTL mapping of 2-propenyl and 3-butenyl GSLs and synteny analysis between LG08 in the present study and A08 of *B. juncea* and *B. rapa*.** The QTLs for 2-propenyl (SIN, Sinigrin) co-localized with the QTLs for 3-butenyl (GNA, Gluconapin) GSL content. 1-LOD and 2-LOD supporting intervals of each QTL were marked by thick and thin bars, respectively. Eight *A. thaliana* markers prefixed "At" (bold and red) on LG08 show a synteny with A08 of *B. juncea* in publish papers [32]. Eighteen ILP markers prefixed "Bnap and Brap" (bold and blue) on LG08 show a

synteny between LG08 and A08 of *B.rapa* through balsting analysis with Brapa_1.0 of *Brassica rapa* cultivar Chiifu in NCBI (https://www.ncbi.nlm.nih.gov/).

## Joint QTL mapping and RNA-sequencing analyses reveal the candidate genes for GSL metabolism in *B. juncea*

To integrate the results of QTL mapping and RNA-sequencing, we performs an alignment analysis between the 24 DEGs related to GSL metabolism and the reference genome of *B. juncea* [16] by the BLAST-like alignment tool [54]. Only two DEGs of LOC106429668 encoding *MYB28/MYB29/MYB76* (physical position: 23,048,306) and LOC106416451 (physical position: 5,833,626) encoding *IIL1* were located on A08 of *B. juncea* genome. In addition, the DNA sequence designed for the PIP markers on A08 also blast the *B. juncea* reference genome [16]. The QTLs for 2-propenyl and 3-Butenyl GSL is located between the physical position of 18,549,777 (BnapPIP592) and the start point of the chromosome, the region of which was overlapped with the position of LOC106416451.

## Discussion

Recombinant inbred lines (RILs) are an important resource in the genetic mapping of complex traits in many species. The RIL mapping population was successfully produced in our laboratory, allowing us to construct a genetic linkage map by utilizing ILP markers in *B. juncea*. The ILP primers were designed on the conserved exons flanking the target intron of cDNA/EST sequences to exploit its polymorphism. Each ILP marker locus might represent one gene copy in the studied genome. Taking the polymorphic and monomorphic loci together, approximately 46.8% of the 306 polymorphic ILP primers in the present study revealed more than one locus, indicating a very close ratio revealed by ILP and RFLP primers in the earlier studies [55,56]. The multiple loci revealed by ILP primers confirmed the polyploidy of *B. juncea*. The higher polymorphism ratio of 24.1% in the ILP primers between the parental lines G266 and G302 not only revealed the hypervariability of ILP primers but also suggested a high degree of variation between the parental genomes. This high genetic difference between the parental

**Table 4. The candidate genes involved in biosynthesis of and transporting of aliphatic GSLs.**

| SN[a] | Gene Name | AGI code | Unigene ID from RNA-Seq | Function |
|---|---|---|---|---|
| 1 | CYP83A1 | AT4G13770 | LOC106447562(U)[b];LOC106391682(U) | aldoxime→s-alkyl-thiohydroximate [46–48] |
| 2 | SOT18 | AT1G74090 | LOC106366617(D);LOC106354324(D);LOC106436726(D) | PAPS-dependent sulfation of desulfo-GSLs→GSLs [49] |
| 3 | SOT17 | AT1G18590 | | |
| 4 | IIL1 | AT4G13430 | LOC106416451(U) | 2-Alkyl-malic acid→3-Alkyl-malic acid [47,50] |
| 5 | IPMI2 | AT2G43100 | LOC106434491(U) | |
| 6 | IPMI SSU1 | AT2G43090 | | |
| 7 | AOP3 | AT4G03050 | LOC106430050(U);LOC106438719(U);LOC106389979(U) | methylsulfinylalkyl GSL→ hydroxyalkyl GSL [47] |
| 8 | SUR1 | AT2G20610 | LOC106440999(D) | s-alkyl-thiohydroximate→ thiohydroximate [47,51,52] |
| 9 | MYB28 | AT5G61420 | LOC106382207(U);LOC106429668(U) | the whole process of biosynthesis of methionine-derived GSL [46] |
| 10 | MYB29 | AT5G07690 | | |
| 11 | MYB76 | AT5G07700 | | |
| 12 | GTR2 | AT5G62680 | LOC106411192(U);LOC106347844(D) | GSL transporting [53] |
| 13 | GTR1 | AT3G47960 | | |

[a]: Serial Number.

[b]: "U" means up-regulated, "D" means down-regulated.

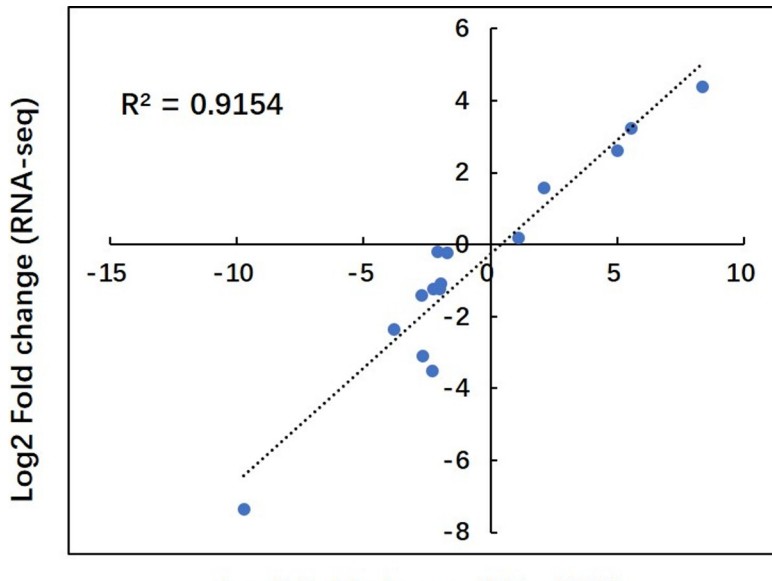

**Fig 5. Comparison of gene expression values obtained by RNA-seq and qRT-PCR.** Fold changes were calculated for 15 DEGs and a high correlation ($R^2 = 0.92$) was observed between the results obtained using the two techniques.

lines would be convenient to increase the density of genetic markers in different linkage groups in *B. juncea*.

The 2-propenyl and 3-butenyl GSLs are the major glucosinolates found in *B. juncea* [57]. The colocated QTL regions of 2-propenyl and 3-butenyl GSLs on A08 of *B. juncea* represented one novel QTL first detected on A08 in Brassiceae [57–59] that explained average phenotypic variations of 42.5% and 42.6%, respectively. The first reason might be that fewer studies focused on 2-propenyl and 3-butenyl GSLs of the germplasm originating in China as the important center of origin in *B. juncea*. Second, the exact order of linkage groups was difficult to obtain before the *B. juncea* genome was sequenced [16]. Another reason might be recombination events between the chromosomes of *B. juncea* in different regions. Although QTL mapping of 2-propenyl and 3-butenyl GSLs has been performed previously in *B. juncea* [57–59], no convenient and reliable primers could be used for gel detection in marker-assisted breeding. In the present study, all 306 primers mapped on *B. juncea* were ILP-type, providing a convenient, specific and rapid detection method for agarose gel electrophoresis in marker-assisted breeding. The ILP marker At4g20150-1 on A08 had the nearest distance of 1.37–3.37 cM to the peak. The ILP primer At4g20150 produced six fragments between 400 bp and 700 bp in the present study and mostly represented six copies of one gene, among which two copies were polymorphic and mapped to LG01 (At4g20150-2) and LG08 (At4g20150-1), respectively. In another genetic map of *B. juncea*, the primer amplified three copies from A3, B7 and B8 [32]. The specific marker of At4g20150 was codominant, clear and simple to score by agarose gel, which resulted in one 700 bp fragment. The novel QTL and the linked marker of At4g20150-1 can be helpful in exploiting the metabolic mechanism of 2-propenyl and 3-butenyl GSLs. The marker tightly linked to QTLs can also be used for marker-assisted selection (MAS). For example, Xu et al. (2018) transferred a thermostable β-amylase from wild barley into a commercial variety, and identified several elite lines with MAS [60], and a major QTL for resistance to Fusarium head blight was transferred from *Thinopyrum elongatum* onto durum wheat 7AL chromosome arm by MAS [61].

GSL synthesis in seeds is nearly nonexistent, so these compounds are mainly imported from other tissues [17,62–64]. As the closest GSL source and the only organ with similar types of GSLs, the siliques might be the source of most seed GSLs in Brassicaceae. In the present study, the RNA-Seq technique was used to screen the key genes for aliphatic GSL synthesis in the siliques of the parental lines G266 and G302. In the siliques of G302, which has high aliphatic GSL contents, 9 DEGs associated with *CYP83A1*, *IIL1*, *IPMI2*, *IPMI SSU1*, *AOP3*, *MYB28*, *MYB29* and *MYB76* were upregulated, resulting in its high GSL contents. Surprisingly, *SUR1* and *SOT17-18* were downregulated for unknown reasons. Furthermore, *GTR2* is upregulated in the siliques of G302, and thus, it might play a more important role in GSL transport from siliques to seeds than the downregulated gene *GTR1*. The process of GSL metabolism is complicated, and few advances have been achieved. The main reasons might be the transport of GSLs among different organs. The combined method of QTL mapping and RNA-Seq should be helpful for the future fine mapping and gene cloning of 2-propenyl and 3-butenyl GSLs.

The joint QTL mapping and RNA-sequencing analyses reveal one candidate gene of LOC106416451 encoding *IIL1*. *IIL1* is mainly responsible for the isomerization of 2-alkyl-malic acid to form 3-alkyl-malic acid [47,50,65,66]. In the present study, *IIL1* is significantly highly (Log$_2$ Fold Change = 9.71, p = 3.58E-15) expressed in the siliques of G302 with high aliphatic GSLs than that in G266 with low aliphatic GSLs. The primary work validates that the *IIL1* might be the key gene for GSL regulation in the present RIL mapping population. However, more work is needed to narrow the QTL region and validate the candidate gene of LOC106416451 in our future study.

In addition, the mapping population used in the present study displayed great variation in agronomic and quality traits in this study and our earlier study [33]. The constructed genetic map would be useful in QTL mapping, gene cloning and marker-based precision breeding of more important traits in *B. juncea*. Because the number of traditional genetic markers is limited, we would sequence the RILs to develop more SNPs (single nucleotide polymorphism) and create a unified, saturated genetic map of *B. juncea* in the future.

## Supporting information

**S1 Table. The synteny analysis between unique transcript of ILP primers and the Brapa_1.0 genome.**
(DOCX)

**S2 Table. The candidate genes involved in biosynthesis of and transporting of aliphatic GSLs.**
(DOCX)

**S3 Table. The 324 transcripts involved in GSL metabolism.**
(XLSX)

**S4 Table. The candidate genes involved in biosynthesis of indolic GLS.**
(DOCX)

**S5 Table. The primers for qRT-PCR validation.**
(DOCX)

## Acknowledgments

The authors thank Shuchun Lin, the agronomist of the teaching and experiment farmer of Guizhou University, for his hard work in field experiment.

## Author Contributions

**Conceptualization:** Kunjiang Yu.

**Investigation:** Aimal Nawaz Khattak, Tianya Wang, Wei Wan, Botao Ye.

**Resources:** Entang Tian.

**Supervision:** Entang Tian.

**Writing – original draft:** Aimal Nawaz Khattak, Tianya Wang, Entang Tian.

**Writing – review & editing:** Aimal Nawaz Khattak, Tianya Wang, Kunjiang Yu, Renqin Yang, Wei Wan, Botao Ye, Entang Tian.

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
