## [Decision Letter · Decision Letter 0]

29 Aug 2019

[EXSCINDED]

PONE-D-19-20230

Exploring the basis of 2-propenyl and 3-butenyl glucosinolate synthesis by QTL mapping and RNA-sequencing in Brassica juncea

PLOS ONE

Dear Mr. Tian,

Thank you for submitting your manuscript to PLOS ONE. After careful consideration, we feel that it has merit but does not fully meet PLOS ONE’s publication criteria as it currently stands. Therefore, we invite you to submit a revised version of the manuscript that addresses the points raised during the review process.

We would appreciate receiving your revised manuscript by Oct 13 2019 11:59PM. To enhance the reproducibility of your results, we recommend that if applicable you deposit your laboratory protocols in protocols.io, where a protocol can be assigned its own identifier (DOI) such that it can be cited independently in the future. For instructions see: http://journals.plos.org/plosone/s/submission-guidelines#loc-laboratory-protocols

We look forward to receiving your revised manuscript.

Kind regards,

Maoteng Li

Academic Editor

PLOS ONE

Journal Requirements:

Reviewers' comments:

Reviewer's Responses to Questions

**Comments to the Author**

1. Is the manuscript technically sound, and do the data support the conclusions?

Reviewer #1: Yes

Reviewer #2: Yes

2. Has the statistical analysis been performed appropriately and rigorously? 

Reviewer #1: Yes

Reviewer #2: Yes

3. Have the authors made all data underlying the findings in their manuscript fully available?

Reviewer #1: Yes

Reviewer #2: Yes

4. Is the manuscript presented in an intelligible fashion and written in standard English?

Reviewer #1: Yes

Reviewer #2: Yes

5. Review Comments to the Author

Reviewer #1: This QTL work is interesting. However, the author didn`t perform a good combination between QTL and RNA-Seq data. Are there some DEGs location in the QTL regions, or all these DEGs are the downstream genes of QTL candidate gene? A small question, DEseq2 uses read count to calculate DEG instead of FPKM.

Reviewer #2: This manuscript constructed a genetic linkage map in B. juncea. Meanwhile, QTL mapping and RNA-seq were used to reveal the 2-propenyl and 3-butenyl GSLs synthesis. The results will be helpful for further fine mapping, gene cloning and genetic mechanisms of 2-propenyl and 3-butenyl GSLs in B. Juncea, which worth publishing in the PLOS one. But still some aspects need to be improved or considered in the further work：

1．Polymorphism comparison between the parental lines, makers, genetic map can be simply introduced in introduction and results; part;

2. The whole genome of B. juncea has been sequenced, how about the synteny relationships between the linkage map and sequencing results? It is better to show some comparative results because different B. juncea lines may have structural variations in certain regions on their genomes.

3. Transcriptome data analysis and processing can be briefly introduced, but with emphasis on describing transcriptome results and the differential expressed genes associated with those QTL results, especially those differential expressed genes involved into the 2-propenyl and 3-butenyl GSLs biosynthetic pathways;

4. The discussion part may be focused on the significance and application of the new QTL detected for breeding. How about the relationship between the QTL on LG08 and previous published ones?

6. PLOS authors have the option to publish the peer review history of their article (what does this mean?). If published, this will include your full peer review and any attached files.

Reviewer #1: No

Reviewer #2: Yes: Yingfen Jiang

---

## [Author Response · Author response to Decision Letter 0]

10 Sep 2019

Maoteng Li

Academic Editor

PLOS ONE

 September 10, 2019

PONE-D-19-20230

Exploring the basis of 2-propenyl and 3-butenyl glucosinolate synthesis by QTL mapping and RNA-sequencing in Brassica juncea

Dear professor Li,

Thank you for your email dated August 29, 2019 regarding our manuscript “Exploring the basis of 2-propenyl and 3-butenyl glucosinolate synthesis by QTL mapping and RNA-sequencing in Brassica juncea submitted to PLOS ONE.” We have accepted the viewers’ comments and revised the manuscript accordingly. Below, we have addressed those comments item by item.

1. To meet PLOS ONE’s style requirements, we have made some revisions as follows:

(1) Line 5 in the revised version – “¶” is used for the co-first author with equal contributions.

(2) Lines 12-18 in the revised version – “These authors contributed equally to this work” is moved to the position just after the e-mail address.

(3) Lines 14-15 in the revised version – “Telephone: +86-851-3855894 and Facsimile: +86-851-83621956” is deleted.

(4) Line 16 in the revised version – “address” is deleted and “(ET)” is added.

(5) Lines 42, 67, 128, 215, 394, 468, 479, 483 and 698 in the revised version –using Level 1 Heading of “Bold type and 18bp font”.

(6) Lines 129, 141, 153, 161, 168, 179-180, 197, 216, 230, 259-260, 321-322 and 335-336 in the revised version –using Level 2 Heading of “Bold type and 16bp font”.

(7) Line 233 in the revised version – “Figs 1 and 2” replaced “Figure 1 and Figure 2”.

(8) Lines 236-238, 280-283, 302-308 and 362-366 in the revised version – “Table 1”, “Table 2”, “Table 3” and “Table 4” are moved into, respectively, from Lines 677-695.

(9) Line 263 in the revised version – “Fig 3” replaced “Figure 3”.

(10) Lines 294, 297, 299 and 328 in the revised version – “Fig 4” replaced “Figure 4”.

(11) Line 332 in the revised version – “S1 Table” replaced “Supplementary S1”.

(12) Line 349 in the revised version – “S3 Table” replaced “Supplementary S2”.

(13) Line 360 in the revised version – “S4 Table” replaced “Supplementary S3”.

(14) Line 372 in the revised version – “S5 Table” replaced “Supplementary S4”.

(15) Line 373 in the revised version – “Fig 5” replaced “Figure 5”.

(16) Lines 469-477 in the revised version – the funding information of “This work was funded by the National Natural Science Foundation of China (Grant No. 31560422), Agricultural Science and Technology Support Program of Guizhou Province (Qiankehe zhicheng No. [2019]2396), Science and Technology Foundation of Guizhou Province of China (Grant No. Qiankehe J zi [2015]2052), Scientific Research Foundation for Returned Scholars, Ministry of Education of China (Grant No. Jiaowaisiliu [2015]1098), Foundation of Guizhou University (Grant No. Guidarenjihezi [2014]14), Construction Program of Biology First-class Discipline in Guizhou (Grant No. GNYL[2017]009).” is deleted, and “The authors thank Shuchun Lin, the agronomist of the teaching and experiment farmer of Guizhou University, for his hard work in field experiment.” is added. 

(17) Lines 698-710 in the revised version – “Supporting information” for S1 Table, S2 Table, S3 Table and S4 Table is added.

2. Response to Reviewer #1:

(1) However, the author didn`t perform a good combination between QTL and RNA-Seq data. 

Reply: We accepted the reviewer’s comment and modified the sentence annotated on the original MS.

Lines 60-61 in the revised version – “Joint QTL mapping and RNA-sequencing analyses reveal one candidate gene of IIL1 (LOC106416451) for GSL metabolism in B. juncea.” is added in the “Abstract” for response to the joint analysis between QTL and RNA-Seq data.

Lines 380-391 in the revised version – “Joint QTL mapping and RNA-sequencing analyses reveal the candidate genes for GSL metabolism in B. juncea. To integrate the results of QTL mapping and RNA-sequencing, we perform an alignment analysis between the 24 DEGs related to GSL metabolism and the reference genome of B. juncea [16] by the BLAST-like alignment tool [54]. Only two DEGs of LOC106429668 encoding MYB28/MYB29/MYB76 (physical position: 23,048,306) and LOC106416451 (physical position: 5,833,626) encoding IIL1 were located on A08 of B. juncea genome. In addition, the DNA sequence designed for the PIP markers on A08 also blast the B. juncea reference genome [16]. The QTLs for 2-propenyl and 3-Butenyl GSL is located between the physical position of 18,549,777 (BnapPIP592) and the start point of the chromosome, the region of which was overlapped with the position of LOC106416451.” is added in the “Results” to perform a combination between QTL and RNA-Seq data.

Lines 452-459 in the revised version – “The joint QTL mapping and RNA-sequencing analyses reveal one candidate gene of LOC106416451 encoding IIL1. IIL1 is mainly responsible for the isomerization of 2-alkyl-malic acid to form 3-alkyl-malic acid [47,50,63,64]. In the present study, IIL1 is significantly highly (Log2 Fold Change=9.71, p=3.58E-15) expressed in the siliques of G302 with high aliphatic GSLs than that in G266 with low aliphatic GSLs. The primary work validates that the IIL1 might be the key gene for GSL regulation in the present RIL mapping population. However, more work is needed to narrow the QTL region and validate the candidate gene of LOC106416451 in our future study.” is added in the “Discussion” to discuss the combination between QTL and RNA-Seq data.

(2) Are there some DEGs location in the QTL regions, or all these DEGs are the downstream genes of QTL candidate gene? 

Reply: We accepted the reviewer’s comment and modified the sentence annotated on the original MS.

Lines 380-391 in the revised version – the gene of LOC106416451 encoding IIL1 is located on the QTL region. 

(3) A small question, DEseq2 uses read count to calculate DEG instead of FPKM.

Reply: We accepted the reviewer’s comment and modified the sentence annotated on the original MS.

Lines 350-356 in the revised version – “To identify differentially expressed genes (DEGs) involved in the GSL metabolism of siliques between G266 and G302. A rigorous algorithm with a threshold of “FDR≤0.05 and │Log2Ratio│≥1” was developed and used as thresholds to judge the significance of differences in transcript abundance.” replaced the original sentence to change the mistake.

3. Response to Reviewer #2:

(1) Polymorphism comparison between the parental lines, makers, genetic map can be simply introduced in introduction and results; part;

Reply: We accepted the reviewer’s comment and modified the sentence annotated on the original MS.

Lines 221-223 in the revised version– “These 306 polymorphic primers consisted of 130 primers from A. thaliana (42.5%), 143 primers from B. napus (19.2%) and 33 primers from B. rapa (13.6%).” was deleted for simplifying the results. 

(2) The whole genome of B. juncea has been sequenced, how about the synteny relationships between the linkage map and sequencing results? It is better to show some comparative results because different B. juncea lines may have structural variations in certain regions on their genomes.

Reply: We accepted the reviewer’s comment and modified the sentence annotated on the original MS.

An excellent suggestion! We can focus on the structural variations of the Brassica juncea genome using one high-density map by SLAF technique in our future work. While we could only primarily use the map to do a primary QTL mapping for its limited number of ILP markers in the genetic map. In addition, we indeed detected the structural variation of the B. juncea genome, giving an example of the synteny analysis of LG08 with one published genetic map of B. juncea as shown in Lines 321-328.

(3) Transcriptome data analysis and processing can be briefly introduced, but with emphasis on describing transcriptome results and the differential expressed genes associated with those QTL results, especially those differential expressed genes involved into the 2-propenyl and 3-butenyl GSLs biosynthetic pathways;

Reply: We accepted the reviewer’s comment and modified the sentence annotated on the original MS.

Lines 60-61 in the revised version – “Joint QTL mapping and RNA-sequencing analyses reveal one candidate gene of IIL1 (LOC106416451) for GSL metabolism in B. juncea.” is added in the “Abstract” for response to the joint analysis between QTL and RNA-Seq data.

Line 340 and 342 in the revised version– “S2 Table” replaced “Table 4” for simplifying the results.

Line 358 in the revised version– “Table 4” replaced “Table 5”.

Lines 380-391 in the revised version – “Joint QTL mapping and RNA-sequencing analyses reveal the candidate genes for GSL metabolism in B. juncea. To integrate the results of QTL mapping and RNA-sequencing, we performs an alignment analysis between the 24 DEGs related to GSL metabolism and the reference genome of B. juncea [16] by the BLAST-like alignment tool [54]. Only two DEGs of LOC106429668 encoding MYB28/MYB29/MYB76 (physical position: 23,048,306) and LOC106416451 (physical position: 5,833,626) encoding IIL1 were located on A08 of B. juncea genome. In addition, the DNA sequence designed for the PIP markers on A08 also blast the B. juncea reference genome [16]. The QTLs for 2-propenyl and 3-Butenyl GSL is located between the physical position of 18,549,777 (BnapPIP592) and the start point of the chromosome, the region of which was overlapped with the position of LOC106416451.” is added in the “Results” to perform a combination between QTL and RNA-Seq data.

Lines 452-459 in the revised version – “The joint QTL mapping and RNA-sequencing analyses reveal one candidate gene of LOC106416451 encoding IIL1. IIL1 is mainly responsible for the isomerization of 2-alkyl-malic acid to form 3-alkyl-malic acid [47,50,63,64]. In the present study, IIL1 is significantly highly (Log2 Fold Change=9.71, p=3.58E-15) expressed in the siliques of G302 with high aliphatic GSLs than that in G266 with low aliphatic GSLs. The primary work validates that the IIL1 might be the key gene for GSL regulation in the present RIL mapping population. However, more work is needed to narrow the QTL region and validate the candidate gene of LOC106416451 in our future study.” is added in the “Discussion” to discuss the combination between QTL and RNA-Seq data.

(4) The discussion part may be focused on the significance and application of the new QTL detected for breeding. How about the relationship between the QTL on LG08 and previous published ones?

Reply: We accepted the reviewer’s comment and modified the sentence annotated on the original MS.

Lines 106-114 and 410-412 in the revised version–we have compared the QTL on LG08 with previous published papers.

Lines 430-436 in the revised version– “The novel QTL and the linked marker of At4g20150-1 can be helpful in exploiting the metabolic mechanism of 2-propenyl and 3-butenyl GSLs and molecular breeding in B. juncea.” was added for the “significance and application of the new QTL detected for breeding.”

Thank you in advance for kind attention!

Sincerely, 

Entang Tian 

Associate professor

Agricultural College of Guihzou University

Telephone: +86-851-3855894

Facsimile: +86-851-83621956

West Campus of Guizhou University

Guiyang City, China

550025

erictian121@163.com

---

## [Decision Letter · Decision Letter 1]

7 Oct 2019

Exploring the basis of 2-propenyl and 3-butenyl glucosinolate synthesis by QTL mapping and RNA-sequencing in Brassica juncea

PONE-D-19-20230R1

Dear Dr. Tian,

We are pleased to inform you that your manuscript has been judged scientifically suitable for publication and will be formally accepted for publication once it complies with all outstanding technical requirements.

With kind regards,

Maoteng Li

Academic Editor

PLOS ONE

Additional Editor Comments (optional):

Reviewers' comments:

Reviewer's Responses to Questions

**Comments to the Author**

1. If the authors have adequately addressed your comments raised in a previous round of review and you feel that this manuscript is now acceptable for publication, you may indicate that here to bypass the “Comments to the Author” section, enter your conflict of interest statement in the “Confidential to Editor” section, and submit your "Accept" recommendation.

Reviewer #1: All comments have been addressed

Reviewer #2: All comments have been addressed

2. Is the manuscript technically sound, and do the data support the conclusions?

Reviewer #1: Yes

Reviewer #2: Yes

3. Has the statistical analysis been performed appropriately and rigorously? 

Reviewer #1: Yes

Reviewer #2: Yes

4. Have the authors made all data underlying the findings in their manuscript fully available?

Reviewer #1: Yes

Reviewer #2: Yes

5. Is the manuscript presented in an intelligible fashion and written in standard English?

Reviewer #1: Yes

Reviewer #2: Yes

6. Review Comments to the Author

Reviewer #1: The authors have answered all my questions and also made necessary revsion. I am satisfy with the revision

Reviewer #2: Authors has addressed all the corcerns raised by reviewers.

I am suggesting that this manuscript of the current version should be accepted for publishing.

7. PLOS authors have the option to publish the peer review history of their article (what does this mean?). If published, this will include your full peer review and any attached files.

Reviewer #1: No

Reviewer #2: No

---

## [Editor Report · Acceptance letter]

10 Oct 2019

PONE-D-19-20230R1 

Exploring the basis of 2-propenyl and 3-butenyl glucosinolate synthesis by QTL mapping and RNA-sequencing in *Brassica juncea*

Dear Dr. Tian:

I am pleased to inform you that your manuscript has been deemed suitable for publication in PLOS ONE. Congratulations! Your manuscript is now with our production department. 

With kind regards,

on behalf of

Dr. Maoteng Li 

Academic Editor

PLOS ONE